# Ultrafast fluxional exchange dynamics in electrolyte solvation sheath of lithium ion battery

Kyung-Koo Lee[1], Kwanghee Park[2,3], Hochan Lee[2,3], Yohan Noh[2], Dorota Kossowska[2,3], Kyungwon Kwak[2,3] & Minhaeng Cho[2,3]

Lithium cation is the charge carrier in lithium-ion battery. Electrolyte solution in lithium-ion battery is usually based on mixed solvents consisting of polar carbonates with different aliphatic chains. Despite various experimental evidences indicating that lithium ion forms a rigid and stable solvation sheath through electrostatic interactions with polar carbonates, both the lithium solvation structure and more importantly fluctuation dynamics and functional role of carbonate solvent molecules have not been fully elucidated yet with femtosecond vibrational spectroscopic methods. Here we investigate the ultrafast carbonate solvent exchange dynamics around lithium ions in electrolyte solutions with coherent two-dimensional infrared spectroscopy and find that the time constants of the formation and dissociation of lithium-ion$\cdots$carbonate complex in solvation sheaths are on a picosecond timescale. We anticipate that such ultrafast microscopic fluxional processes in lithium-solvent complexes could provide an important clue to understanding macroscopic mobility of lithium cation in lithium-ion battery on a molecular level.

[1] Department of Chemistry, Kunsan National University, Kunsan, Jeonbuk 573-701, Korea. [2] Center for Molecular Spectroscopy and Dynamics, Institute for Basic Science (IBS), Korea University, Seoul 02841, Korea. [3] Department of Chemistry, Korea University, Seoul 02841, Korea. Correspondence and requests for materials should be addressed to K.K. (email: kkwak@korea.ac.kr) or to M.C. (email: mcho@korea.ac.kr).

Electrolyte solvent capable of dissolving lithium salt is an integral part of Li-ion battery (LIB) that has become an indispensable power source for current mobile ecosystem and electrical vehicles[1–4]. Among various components in LIB, the properties and dynamics of carbonate-based solvents govern the mobility and stability of charge carrier, $Li^+$ cation, in LIB[5,6]. Because $Li^+$ ion mobility is much lower than electrons in metal wires, the solvation dynamics of $Li^+$ in electrolyte solutions is crucial in determining the LIB performance. Typical electrolyte solutions used in LIB are composed of non-aqueous carbonate solvents that have certain advantages. For instance, the oxidation potential can be matched to the output voltage of LIB and carbonate solvents are also building blocks of solid-electrolyte interphase[7–11].

To achieve high $Li^+$ conductivity, mixed solvents have often been used not only because they increase the solubility of lithium salt but also because they serve as excellent media facilitating fast transport of Li-ion[6]. Cyclic carbonates, such as propylene carbonate (PC) and ethylene carbonate (EC), have high dielectric constants so that they enhance the dissociation of lithium salt by stabilizing ions. However, their high viscosity limits the fast $Li^+$ transport. To enhance $Li^+$ ion mobility as well as to improve low-temperature performance, linear alkyl carbonates like diethyl carbonate (DEC) and dimethyl carbonate (DMC) are added to the LIB electrolyte solutions as co-solvents. It has long been believed that such low viscosity co-solvents act as a medium for transporting the sheaths of $Li^+$ ion-solvent complex[12]. In the current commercial LIBs, lithium salt, 1.0 M $LiPF_6$, is dissolved in organic carbonate mixed solvents consisting of equal volumetric amounts of DEC, DMC and EC. $Li^+$ is believed to be preferentially solvated by EC, and DEC and/or DMC provides the medium for large Li–EC complexes to flow between electrodes during charging and discharging processes. However, to our surprise, detailed microscopic solvation structures and ultrafast solvation dynamics around $Li^+$ in such electrolyte solutions have not been fully understood and even they are still the key subjects under intense scholarly debate.

Therefore, elucidating the molecular solvation structure of organic carbonates around $Li^+$ would, without a doubt, help one to better understand the working principles of LIB and to eventually design an improved LIB. Most of the previous studies primarily focused on the equilibrium structure of the first solvation shell of $Li^+$, which has often been referred to as solvation sheath due to its rigidity[13–17]. For instance, Raman spectroscopy of LIB electrolyte solutions revealed the existence of preferential solvation shell around $Li^+$ in mixed solvent electrolyte solutions[18,19]. This was also confirmed by Bogle et al.[20] where [17]O NMR experiments show that $Li^+$ makes a direct electrostatic interaction with carbonyl oxygen atoms of surrounding carbonates. Even in the case that $Li^+$ is preferentially solvated by EC or PC, the participation of co-solvent molecules, such as DEC and DMC, in the solvation sheath is possible not only because of their high-mole fractions but also because of extra entropic gain. Therefore, it is by now clear that such co-solvents could potentially play a dual-role of low-viscous medium and solvating molecule. Co-solvents like DEC and DMC would add an extra flexibility to solvation sheath, because they comparatively weakly interact with $Li^+$ and have many conformational degrees of freedom[21]. Nevertheless, no direct evidence of fast fluxional solvent exchange dynamics in $Li^+$ solvation sheath has been reported due to their ultrafast nature and experimental difficulties in finding an appropriate spectroscopic probe.

In this regard, we believe that time-resolved nonlinear infrared spectroscopy is an ideal method capable of providing critical information on ultrafast chemical exchanges between different solvation structures[22–27] that cannot be easily distinguished by NMR or other spectroscopic means due either to their limited time-resolving power or to a lack of spectroscopic probe enabling one to track local configurational changes around $Li^+$ ion. Here considering the carbonyl stretching vibration of DEC as an infrared probe, we carried out Fourier transform infrared (FTIR), femtosecond infrared pump-probe and two-dimensional (2D) infrared studies of chemical exchange processes between DEC molecules with and without Coulombic interaction with $Li^+$. One of the most difficult problems in the present linear and nonlinear infrared spectroscopic investigations is that the infrared probe, DEC, itself is the solvent too. Thus, with normal infrared cells with several micrometre thickness at best, the linear absorption is saturated and the nonlinear infrared signal fields are self-attenuated, which renders a difficulty in detecting weak 2D infrared signal. To overcome this, we prepared a home-built thin sample cell coated with $SiO_2$, using radio frequency magnetron sputtering technique (see Fig. 1a,b and Supplementary Fig. 1).

Here we show evidence on the ultrafast formation and dissociation dynamics of Li–carbonate solvation complexes in the LIB electrolyte solutions by analysing the FTIR spectra and time-resolved infrared pump-probe and 2D infrared data of $C=O$ stretch modes in lithium salt electrolyte solutions and carrying out quantum chemistry calculations. In particular, the time constant associated with the breaking of $Li^+ \cdots DEC$ electrostatic interaction is found to be 2.2 ps and that with the formation of the $Li^+ \cdots DEC$ bond is 17.5 ps at 1.0 M $LiPF_6$ concentration in DEC solvent. The timescales of making and breaking of $Li^+ \cdots DMC$ interaction bonds in PC/DMC mixed solvents are similar to those for DEC solutions, which indicates that the DMC molecules also participate in the formation of lithium cation solvation shell even in such mixed electrolyte solutions used in a real LIB. We anticipate that such ultrafast fluxional dynamics in solvation shell structures around lithium cations are of importance in understanding lithium-ion mobility in LIB solutions.

## Results

**FTIR spectroscopy.** The infrared absorption spectra of the $C=O$ stretch and $O-C-O$ asymmetric stretch modes in $LiPF_6$ DEC solutions at different concentrations are shown in Fig. 1c,d, respectively. DEC has three strongly infrared-active modes in the fingerprint region, which are $C=O$ stretching ($1,747\ cm^{-1}$), $O-C-O$ asymmetric stretching ($1,258.9\ cm^{-1}$), and $O-CH_2-CH_3$ asymmetric stretching ($1,022\ cm^{-1}$) vibrations. As $LiPF_6$ concentration in the DEC solution increases, two distinctively new peaks at 1,715.4 and $1,305.6\ cm^{-1}$ appear. The frequencies and spectral widths of the main peaks at 1,747 and $1,258.9\ cm^{-1}$ do not change much on increasing lithium salt concentration. Therefore, the two new peaks at 1,715.4 and $1,305.6\ cm^{-1}$ can be safely assigned to the $C=O$ stretch and $O-C-O$ asymmetric stretch modes of DEC molecules directly interacting with $Li^+$.

However, it is noted that a DEC molecule has three different sites for electrostatic interaction with $Li^+$. They are two ester ether oxygen atoms and one carbonyl oxygen atom. Therefore, at least two structures in Fig. 2 should be considered as possible Li–DEC complexes. Despite that a recent [17]O NMR experiment provided evidence that the ester ether oxygen atoms do not strongly interact with Li-ion[20], it is not entirely clear whether this is the case even at high (1.0 M) lithium salt concentration. To pin point the Li–DEC complex structure among the two in Fig. 2, quantum chemistry calculations and vibrational analyses of

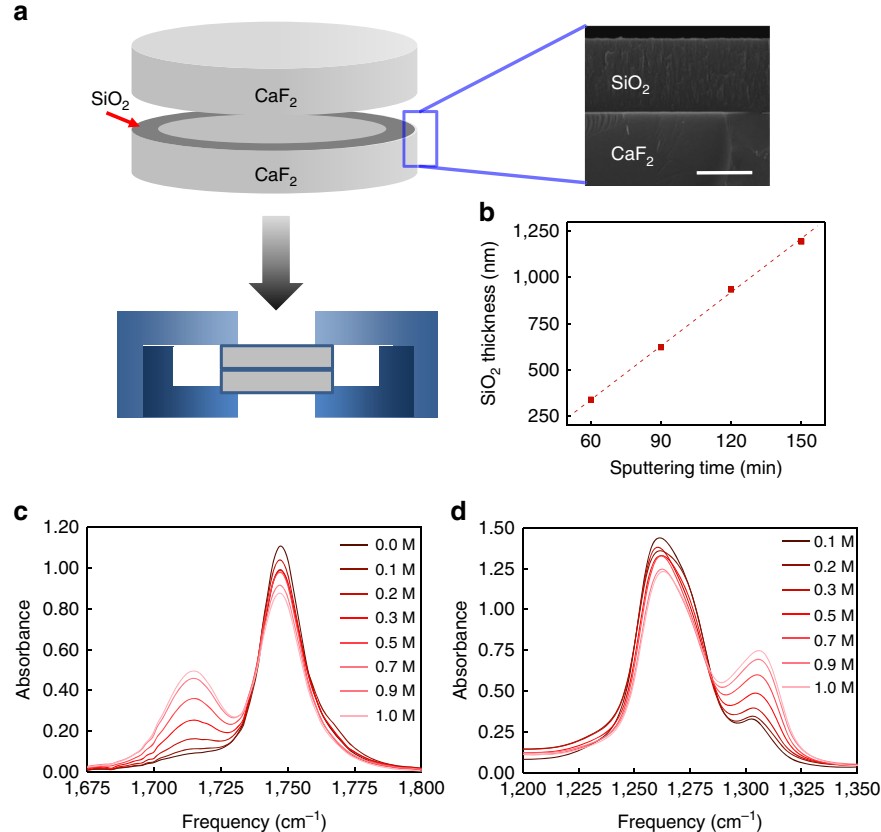

**Figure 1 | Home-built thin infrared sample cell and FTIR spectra. (a)** RF magnetron sputtering method was used to deposit a thin donut-shape $SiO_2$ film on $CaF_2$. The thickness of $SiO_2$ film on $CaF_2$ was measured with scanning electron microscopy (right panel). Scale bar, 500 nm. **(b)** The linear relationship between the sputtering time and the thickness of $SiO_2$ film was used to control the infrared beam path length. **(c)** The FT-IR spectra of the C=O stretch and **(d)** the O–C–O asymmetric stretch of $LiPF_6$ DEC solution at various $LiPF_6$ concentrations.

a few complexes were performed with DFT method at the level of B3LYP/6-311+ +G(3df,2pd). For an isolated DEC, the C=O stretch and O−C−O asymmetric stretch frequencies are found to be 1,714.4 $cm^{-1}$ and 1,235.5 $cm^{-1}$, respectively (detailed descriptions on the quantum chemistry calculation results are found in Supplementary Note 2). Then, the DFT calculations of two structures in Fig. 2 show that the C=O···$Li^+$ complex (Fig. 2a) is more stable by 3.26 kcal $mol^{-1}$ than the O=C−O···$Li^+$ complex in Fig. 2b. Furthermore, when $Li^+$ interacts with carbonyl oxygen atom, the C=O stretch frequency is red shifted by 112.2 $cm^{-1}$ and simultaneously the O−C−O asymmetric stretch frequency is blue shifted by 62.6 $m^{-1}$ (Supplementary Table 1). In contrast, the complex formation of $Li^+$ with two ester ether oxygen atoms of DEC induces strong blue shift (112.4 $cm^{-1}$) of C=O frequency and red shift (96.8 $cm^{-1}$) of O−C−O asymmetric stretch frequency. These results can be understood by noting that the $Li^+$ ion interacting with carbonyl oxygen atom makes the double bond character (or bond length) of C=O group decrease (increase), which concomitantly makes the C−O bond lengths short (Supplementary Table 1). When $Li^+$ electrostatically interacts with ester ether oxygen atom, exactly the opposite charge redistribution occurs. Thus, our infrared results are consistent with the previous notion that $Li^+$ preferentially interacts with carbonyl oxygen atom (Fig. 2a)[28,29]. In fact, such frequency shifting behaviours of C=O and O−C−O modes in DEC on the formation of C=O···$Li^+$ complex are quite similar to the red and blue shifts of amide I and II modes of peptide bond, respectively, which are mainly C=O stretch and C−N stretch of the peptide bond, respectively, when the amide carbonyl oxygen

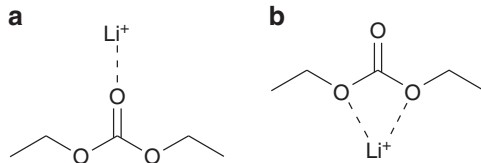

**Figure 2 | Two $Li^+$···DEC dimer structures.** $Li^+$ in DEC solution can interact with **(a)** carbonyl oxygen atom, C=O···$Li^+$, or **(b)** with two ester ether oxygen atoms of DEC molecule, O=C(-O)$_2$:::$Li^+$.

atom forms a hydrogen bond with a water molecule. Despite that the quantum chemistry calculations for the two different complex structures in Fig. 2 are consistent with previous findings and our experimental FT-IR results discussed below, the C=O stretch frequency may depend also on the side chain (ethyl group) conformation. More specifically, the molecular dipole moment of the *cis* conformation can be larger than that of the *trans* conformation so that the *cis* conformer can make a bit stronger interaction with $Li^+$ and form a more stable Li–DEC complex.

The number of solvent molecules in the first solvation shell of a solute molecule is an important quantity providing information on the size and structure of Li-ion solvation sheath. Often, a series of lithium salt concentration-dependent spectroscopic experiments has been used to determine the solvation number around a $Li^+$ in various electrolyte solutions[28,29]. However, Doucey *et al.*[13] showed that the formation of contact

ion pair (CIP) between two oppositely charged ions should also be taken into account in quantitatively interpreting the infrared spectra of LIB solutions. That means, the concentration of dissolved LiPF$_6$ is not necessarily identical to the concentration of DEC-solvated Li$^+$ ion because there could exist non-negligible amount of Li$^+$:PF$_6^-$ CIP. Thus, it is necessary to consider three different DEC species that are (i) DEC surrounded by other DECs, (ii) DEC interacting with free Li$^+$ and (iii) DEC interacting with Li$^+$ in the CIP state, Li$^+$:PF$_6^-$. In addition, the existence of two different types of anions, either PF$_6^-$ solvated by DECs or PF$_6^-$ in CIP imposes yet another complexity to the interpretation of linear and nonlinear infrared spectra. However, because the negatively charged PF$_6^-$ tends to be away from the carbonate part of DEC due to Coulomb repulsion and preferentially surrounded by the DEC ethyl groups[30], PF$_6^-$ does not strongly affect the DEC carbonyl stretch infrared spectrum.

To gain more insight into the vibrational properties of DEC$\cdots$Li$^+$:PF$_6^-$ as compared to DEC$\cdots$Li$^+$, we further analysed density functional theory calculation results of DEC interacting with Li$^+$:PF$_6^-$ (Supplementary Fig. 4). Even in the case of Li$^+$ in Li$^+$:PF$_6^-$, still the carbonyl oxygen atom of DEC is the binding site forming C$=$O$\cdots$Li$^+$:PF$_6^-$ complex, though the C$=$O frequency red shift in that case is a bit smaller than that of C$=$O$\cdots$Li$^+$. If the first solvation shell structure around free Li$^+$ is spectroscopically different from that around Li$^+$:PF$_6^-$, one might expect to see clear signature, that is, a new third infrared absorption band or shoulder in the lower-frequency region. A careful examination of the C$=$O band of LiPF$_6$ DEC solution, where the total C$=$O stretch infrared band was corrected by subtracting that of pure DEC liquid, shows no discernible spectroscopic feature of DEC-CIP (Supplementary Fig. 3). Thus, it is concluded that the carbonyl stretch infrared band consists of indistinguishable contributions from both C$=$O$\cdots$Li$^+$ and C$=$O$\cdots$Li$^+$:PF$_6^-$ complexes, which altogether will be referred to as Li–DEC complex.

**Polarization-controlled infrared pump-probe.** Figure 3a,b displays the dispersive isotropic pump-probe signals obtained from pure DEC liquid and 1.0 M LiPF$_6$ DEC solution, respectively. The time-resolved pump-probe spectra of LiPF$_6$ in DEC solution show complicated spectral features due to the destructive interference between the negative 1–2 transition peak (excited state absorption transition from |1> to |2>) of free DEC and the positive 0–1 peak (ground state |0> bleach and stimulated emission from |1> to |0>) of Li–DEC complex at around 1,720 cm$^{-1}$. Another notable feature is the prolonged positive signal (around 1,750 cm$^{-1}$ in Fig. 3a,b) that is associated with local heating effect due to the vibrational energy relaxation of infrared-pump-excited C$=$O stretch modes[31]. The vibrational lifetime could be extracted from the fitting analysis of the 1–2 transition PP data that are relatively immune to local heating effect. Note that the decay constants of the excited state absorption signals at around 1,725 cm$^{-1}$ and 1,700 cm$^{-1}$, respectively, for pure DEC liquid and 1.0 M LiPF$_6$ DEC solution, are little dependent on the probe frequency (Supplementary Fig. 5). On the other hand, the exponential decay constants obtained from the fits of positive 0–1 signals at $\sim$1,750 cm$^{-1}$ show a strong probe-frequency dependence, which is mainly due to the local heating contribution to the infrared PP signal. Thus, the isotropic and anisotropic data at the 1–2 transition frequencies at $\sim$1,700 cm$^{-1}$ were taken into consideration to extract information on the vibrational and orientational relaxation times of Li–DEC complex. The corresponding time constants of free DEC were obtained

from the excited state absorption signal at 1,725 cm$^{-1}$ in the infrared PP data of pure DEC solvent (Fig. 3a).

Vibrational lifetime of Li–DEC complex (1.1 ± 0.1 ps) is found to be much shorter than that of free DEC (2.1 ± 0.1 ps). Detailed discussion on the origin of such increased vibrational relaxation rate of DEC upon its electrostatic interaction with Li$^+$ can be found in Supplementary Note 3. The anisotropic infrared pump-probe signals of free DEC and Li–DEC enabled us to determine the orientational relaxation rates. As expected, the Li–DEC complex with three or four DEC molecules with or without ion-pairing PF$_6^-$ rotates slowly as compared to free DEC and the second-order rotational correlation time constants are found to be 8.3 ± 0.6 ps and 1.5 ± 0.1 ps for Li–DEC complex and free DEC, respectively.

**2D infrared chemical exchange spectroscopy.** Figure 3c shows four representative time-resolved 2D infrared spectra of C$=$O stretch mode of DEC in 1.0 M LiPF$_6$ DEC solution at different waiting times. Each 2D infrared spectrum was normalized to the largest positive 0–1 peak magnitude. At a very short waiting time of $T_w = 0.3$ ps, there are only two peaks on the diagonal line (see the upper-left panel in Fig. 3c), which correspond to the two absorption peaks in the FTIR spectrum shown in Fig. 1c. The positive peaks (red colour) result from the ground state bleaching and stimulated emission contributions, whereas the negative peaks (blue colour) in the lower frequency region along the probe frequency ($\omega_t$) axis are the excited state absorption contributions. Both positive and negative peaks are elongated along the diagonal line at short waiting time up to $T_w = 1.0$ ps. However, after 2.0 ps, all peaks become round in shape due to the spectral diffusion process. Diagonally elongated peaks at short waiting times indicate structural heterogeneity, distribution of coordination number and ethyl side-chain conformational distribution (cis-cis, cis-trans, and trans-trans)[29] of DEC in solvation complexes around Li$^+$. We found that the C$=$O stretch frequency–frequency correlation decays on a few picosecond timescale, which could result from ultrafast internal rotations of the ethyl groups in DEC and hindered rotational motions of carbonates.

In principle, the inhomogeneous structural distribution can be studied by analysing the centre or nodal line slope and/or diagonal ellipticities of 2D infrared peaks[32–34]. Unfortunately, due to the destructive interference between the positive 0–1 peak from Li–DEC and the negative 1–2 peak from free DEC, it is difficult to clearly identify the corresponding 2D infrared feature. Thus, we compared the infrared absorption linewidths of free DEC and Li–DEC complex with the help of pump-probe data. The FWHM (full-width at half-maximum) of the infrared spectrum of Li–DEC is found to be 22.2 cm$^{-1}$, which is slightly broader than that of free DEC (FWHM = 18.9 cm$^{-1}$). However, the vibrational lifetime of Li–DEC is just half of that of the free DEC, which means that the lifetime broadening of Li–DEC is larger than that of free DEC. These observations indicate that the inhomogeneous linewidth of Li–DEC may be narrower than free DEC, which is consistent with the notion that DEC molecules in Li–DEC complexes have stable conformers and the side-chain structure can be correlated with its solvation property[35].

In a real LIB electrolyte solution, detailed solvation dynamics around Li$^+$ could be affected by the presence of co-solvents in the first solvation shell. In a recent $^{17}$O NMR experiment, NMR peak of carbonyl oxygen atom in EC or DMC shows a motional narrowing behaviour, indicating the existence of fast equilibrium dynamics between free carbonate and Li-complexed carbonate molecules in nanosecond or even shorter timescales[20].

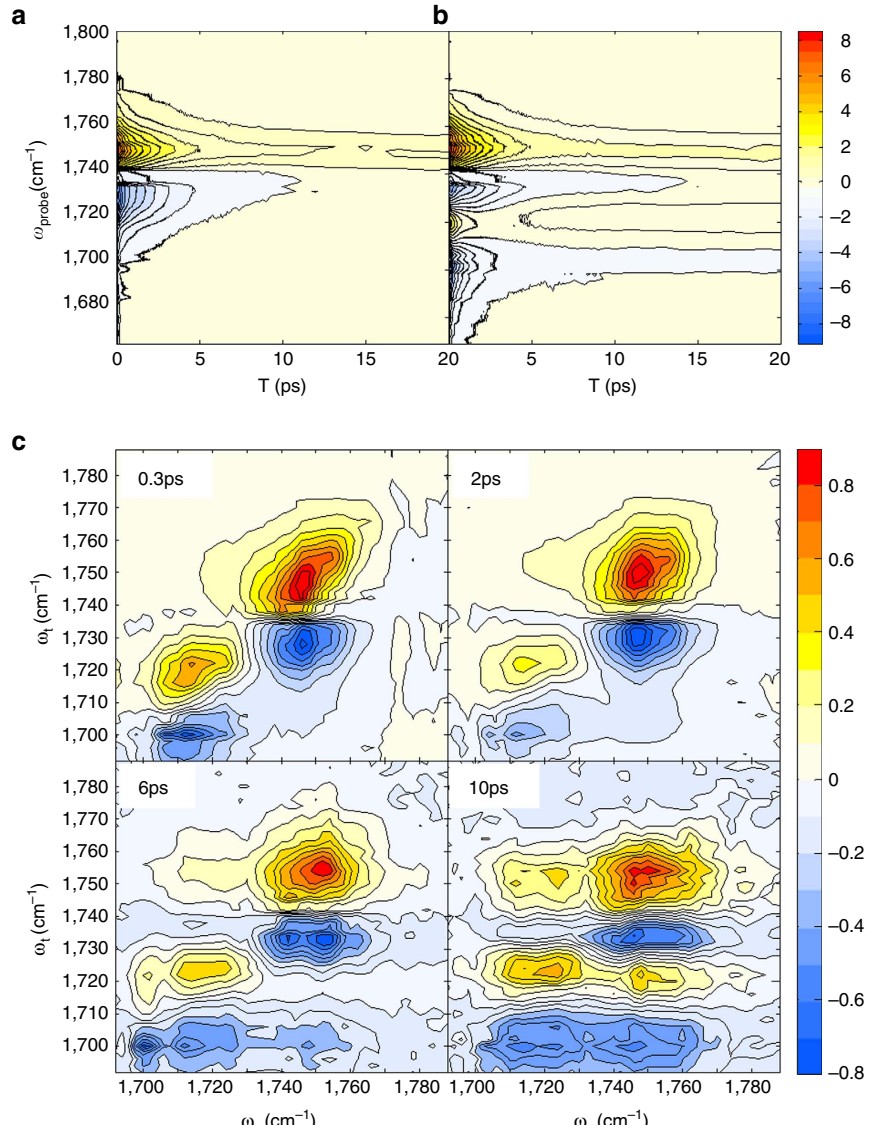

**Figure 3 | Dispersed infrared pump-probe and absorptive 2D infrared spectra.** Time-and-frequency-resolved infrared pump-probe data of pure DEC liquid (**a**) and 1.0 M LiPF$_6$ DEC solution (**b**) are plotted. (**c**) Four 2D infrared spectra of 1.0 M LiPF$_6$ DEC solution at different waiting times are shown, where each 2D infrared spectrum was normalized to the positive peak maximum value. The x and y axes are the infrared frequency of the pump (excitation) and probe (emission) fields, respectively. In these contour plot, red (blue) colour represent positive (negative) sign.

Interestingly, it has long been believed that such fast solvent exchange dynamics can occur only in the second solvation shell because of the belief that the first solvation shell maintains its rigid structure during the Li-transport between electrodes[5,6]. However, if Li-solvent complexes undergo fast solvent exchange dynamics resulting in ultrafast fluctuation of coordination number and solvation structure, the conventional view on the robust and rigid solvation sheath formation must be reconsidered. Especially the coordination number fluctuation would modulate the diffusion rate of Li$^+$ in LIB electrolyte solutions. To date, however, little has been understood about how fast the solvation dynamics and the structural fluctuation of solvation sheath.

In this regard, the 2D infrared spectroscopy is an ideal tool for studying such ultrafast structural changes inside each solvation shell, because it is capable of tracking frequency changes of infrared probe (C=O stretch mode of DEC in the present case) induced by conformational transitions between different solvation structures. Indeed, at longer waiting time ($T_w > 2$ ps), one can

clearly observe cross peaks (Fig. 3c). The positive cross peak on the upper-left corner results from the breaking of Li$^+ \cdots$DEC, that is, transition from Li–DEC to free DEC, which reduces the number of DEC solvent molecules in the first solvation shell (the backward reaction in Fig. 4a). Another positive cross peak appears on the lower-right corner, even though its magnitude is comparatively weak because of its destructive interference with the negative diagonal peak from free DEC. The positive and negative cross peaks in the lower-right corner represent the formation of Li–DEC complex or the increase of coordination number around Li$^+$ ion (the forward reaction in Fig. 4a).

The Li–DEC coordination breaking and forming dynamics are manifested in the $T_w$- dependent cross peak intensity changes on the 2D infrared spectra. As mentioned earlier, there are three different DEC species in electrolyte solutions that are free DEC, DEC$\cdots$Li$^+$(DEC)$_{n-1}$ and DEC$\cdots$Li$^+$:PF$_6^-$(DEC)$_{n-2}$. However, DEC in DEC$\cdots$Li$^+$(DEC)$_{n-1}$ and that in DEC$\cdots$Li$^+$:PF$_6^-$(DEC)$_{n-2}$ are neither spectrally nor kinetically

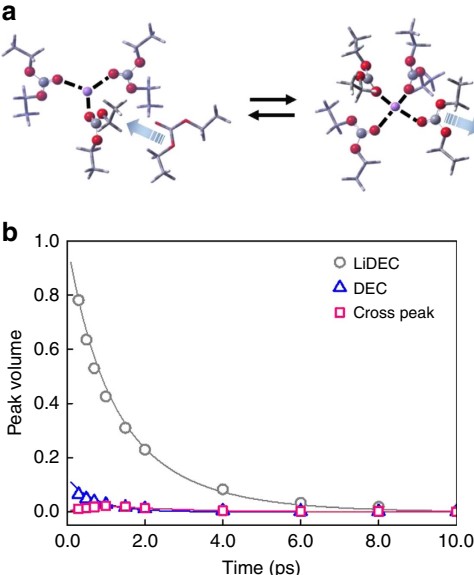

**Figure 4 | Equilibrium structure and kinetic analysis of 2D infrared spectrum.** (**a**) A schematic picture on the chemical equilibrium between two solvation structures. The left consists of Li-complex with three bound DECs and one free DEC, whereas the right represents the tetra-coordinated Li–DEC complex. (**b**) The time-dependent diagonal and cross peak volumes obtained from the experimentally measured 2D infrared spectra are plotted. The fitted (solid) curves are also shown here.

resolvable, due to the similarity in solvation structures. This led us to use a simple two-state exchange model between DEC and DEC$\cdots$Li$^+$(ligands)$_{n-1}$, where ligand refers to either DEC or PF$_6^-$ and $n$ varies depending on lithium salt concentration and solvent composition. Here the chemical exchange dynamics is therefore the dynamical equilibrium between Li$^+$(ligands)$_n$ and Li$^+$(ligands)$_{n-1}$ + free DEC (Fig. 4a).

Equilibrium population ratio between free DEC and DEC$\cdots$Li$^+$(ligands)$_{n-1}$ can be obtained from the infrared spectral peak analyses after extracting information on the corresponding transition dipole ratio from the LiPF$_6$ concentration-dependent infrared data (Supplementary Note 4). The transition dipole strength of DEC$\cdots$Li$^+$(ligands)$_{n-1}$ is 1.7 times larger than that of free DEC, which is in excellent agreement with quantum chemistry calculation results (Supplementary Fig. 4 and Table 2). With the experimentally measured transition dipole ratio of 1.7, the population ratio of DEC$\cdots$Li$^+$(ligands)$_{n-1}$ to DEC is found to be 0.138, that is, [DEC$\cdots$Li$^+$(ligands)$_{n-1}$]/[free DEC] = 0.138. From the peak volumes (Fig. 4b) obtained through a 2D Gaussian fitting analysis of 2D infrared spectrum, the time-dependent concentrations of free DEC and Li–DEC complex were obtained. All the other parameters required in the present volume fitting analyses of 2D infrared data were obtained from the FTIR and infrared pump-probe measurements. The two-state-model fit curves shown in Fig. 4b are in excellent agreement with experimental data. Finally, it is found that the time constant associated with the breaking of Li$^+\cdots$DEC is 2.2 ps and that with the formation of one Li$^+\cdots$DEC bond is 17.5 ps at 1.0 M LiPF$_6$ concentration.

Although our experimental observations clearly show the ultrafast chemical exchange dynamics in lithium-ion solvation sheath, we still need to consider another possibility that the increasing cross peaks could result from vibrational excitation transfers between free DEC and Li–DEC complex. Indeed, the vibration energy transfer between donor and acceptor

molecules with different transition frequencies can produce cross peaks of which intensities increase in time $T_w$ (ref. 36). Unlike the case of chemical exchange-induced cross peaks, the increase rate of the upper-left cross peak intensity should differ from that of the lower-right cross peak intensity, because they are produced by up- and down-hill transitions, respectively.

Unfortunately, the lower-right positive cross peak in Fig. 4b overlaps with the negative diagonal peak from the free DEC, which obscures the time-dependent features of the cross peaks. We thus compared the time-dependent increase of the upper-left positive cross peak and that of the lower-right negative peak, assuming that the chemical exchange dynamics of DECs (either free DEC or Li–DEC complex) in their excited states are similar to those in their ground states. To rule out the possibility that the 2D infrared cross peaks originate from vibrational excitation transfers between free DEC and Li–DEC complex, we further carried out a series of concentration-dependent pump-probe measurements at four different LiPF$_6$ concentrations in the range from 0.2 to 1.0 M (detailed analysis on pump-probe results are found in Supplementary Note 3). A decrease in LiPF$_6$ concentration would significantly reduce the chance for a free DEC to encounter Li-complexed DEC molecules. This should result in an increase of vibrational lifetime of free DEC, if the vibrational energy transfer between free DEC and complex DEC occurs within a few picosecond timescale. However, as shown in Supplementary Fig. 6, the vibrational lifetime of free DEC is independent of LiPF$_6$ concentration. This confirms that the cross peaks observed in our 2D infrared spectra originate from ultrafast chemical exchange processes in the solvation sheath of Li$^+$ in DEC electrolyte solution.

**2D infrared chemical exchange spectroscopy of mixed solvent.** Despite that the ultrafast chemical exchange dynamics between free DEC and Li–DEC complex was observed in pure DEC solutions, typical LIB's use mixed solvents instead. To elucidate the separate roles and solvation properties of co-solvents with DMC and PC and to investigate the possibility of DMC playing dual roles both as a solvent medium for mobile Li–PC(DMC) complexes and as a ligand participating in the first solvation shell around Li$^+$, we carried out 2D infrared chemical exchange measurements for LiPF$_6$ solutions in the more realistic DMC and PC mixed solvents, which are (i) 1.0 M LiPF$_6$ in DMC:PC (1:1 in volume percent) solution, (ii) 1.5 M LiPF$_6$ in DMC:PC (1.5:1) solution and (iii) 2.0 M LiPF$_6$ in DMC:PC (1.5:1) solution. Here it should be noted that we considered DMC:PC mixed solvents instead of DEC:PC mixed solvents because of the following reasons. First, since the C=O stretching vibrational lifetime of DMC is comparatively longer than that of DEC, we could observe measurable diagonal and cross peaks of free DMC and Li–DMC complex in our 2D infrared spectra at long waiting times. Second, we here show that the chemical exchanges observed in Li–DEC (single carbonate solvent) solution do occur in different mixed solvent systems too.

Despite that the 2D infrared spectra appear to be spectrally congested, we could clearly observe real-time chemical exchange processes between Li–DMC$_n$PC$_m$ complex with $n$ DMC and $m$ PC molecules in the first solvation shell and free DMC molecules even in the DMC and PC (1:1 in volume percent) mixed solvents with 1.0 M LiPF$_6$ (see Fig. 5). The 2D infrared spectra for other mixed solvent systems with different solvent compositions are presented in Supplementary Fig. 10. All four representative 2D infrared spectra in Fig. 5 show four diagonal peaks, where the two in the lower frequency region at around 1,725 and 1,760 cm$^{-1}$ are associated with the C=O stretch

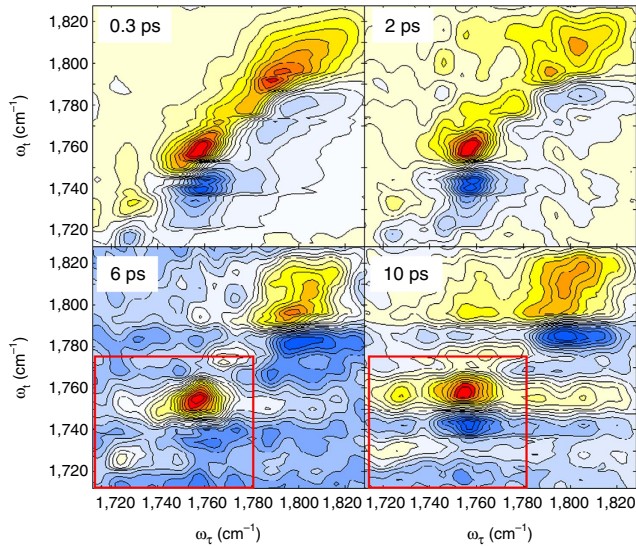

**Figure 5 | Four representative two dimensional IR spectra of 1.0 M LiPF6/DMC:PC (1:1 v/v%) solution at different waiting times.** Each 2D infrared spectrum was normalized to the positive peak maximum value.

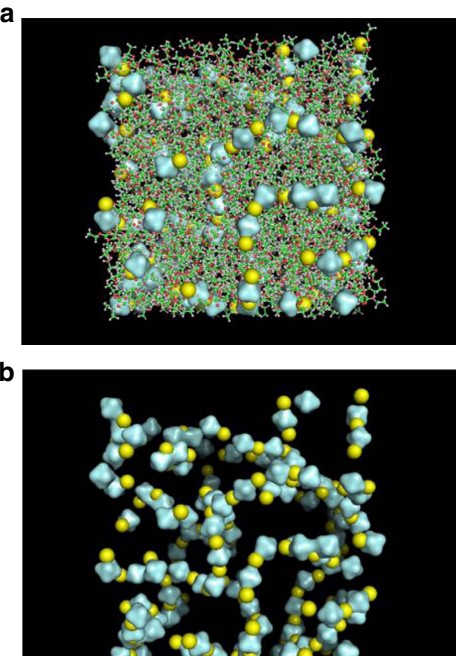

**Figure 6 | Snapshot structure of 1.0 M LiPF₆ DEC solution obtained from MD simulation trajectories.** (**a**) Li⁺ and PF₆⁻ are represented by yellow spheres and grey octagons, respectively. (**b**) The ions in this LIB electrolyte solution form three-dimensionally extended ion network structures.

modes of Li–DMC and free DMC molecules, respectively, and the two in the higher-frequency region at around 1,780 and 1,800 cm⁻¹ originate from the C=O stretch modes of Li–PC and free PC molecules (the FTIR spectra of three different mixed solvent systems (i)–(iii) can be found in Supplementary Fig. 9).

As waiting time increases, the cross peak amplitude at $\omega_\tau = 1{,}725$ cm⁻¹ and $\omega_t = 1{,}760$ cm⁻¹ increases (see the 2D infrared spectra inside the red boxes in Fig. 5), which clearly indicates the chemical exchange or configurational transition from Li–DMC to free DMC. In contrast, we could not find any cross peak features between Li–PC and free PC in our 2D infrared spectra in Fig. 5 and Supplementary Figs 10–12 within our experimental timescale. This suggests that the electrostatic interaction between PC and Li⁺ in LIB electrolyte solutions is stronger than that between DMC (or DEC) and Li⁺, which is consistent with the notion that the PC molecule with large dipole moment compared to that of DEC molecule tends to make a stable and rigid solvation sheath around a lithium cation in LIB electrolyte solutions.

To approximately examine the timescale of chemical exchange processes between Li–DMC complex and free DMC in 1.0 M LiPF₆/DMC:PC (1:1 in volume percent) solution, we fitted the 2D infrared peaks with a 2D Gaussian function (Supplementary Figs 10–12), which allow us to determine the volumes of the two positive diagonal peaks at 1,725 and 1,760 cm⁻¹ and the corresponding positive cross peak. Note that the two diagonal peak intensities of Li–DMC and free DMC at 1,725 cm⁻¹ and 1,760 cm⁻¹, which are denoted as $I_{\text{Li–DMC}}^{\text{diag}}$ and $I_{\text{DMC}}^{\text{diag}}$, respectively, are determined by a few factors that include (i) the transition dipole moments, $|\mu_{\text{Li–DMC}}|^4$ and $|\mu_{\text{DMC}}|^4$, where $\mu_{\text{Li–DMC}}$ and $\mu_{\text{DMC}}$ are the C=O stretching vibrational transition dipole moment of Li–DMC complex and free DMC molecule, (ii) vibrational lifetimes, (iii) rotational relaxation times and (iv) survival probabilities of the two species within the waiting time. The cross peak intensity, denoted as $I_{\text{LiDMC:DMC}}^{\text{cross}}$, between Li–DMC and free DMC is determined by (i) the transition dipole moment, $|\mu_{\text{Li–DMC}}|^2|\mu_{\text{DMC}}|^2$, (ii) vibrational lifetimes, (iii) rotational relaxation times and (iv) condition probability of finding free DMC after finite waiting time when it was initially in a Li–DMC complex form. Therefore, the ratio of the cross peak intensity to the square root of the product of the two diagonal peaks, that is, $I_{\text{Li–DMC:DMC}}^{\text{cross}} / \sqrt{I_{\text{Li–DMC}}^{\text{diag}} I_{\text{DMC}}^{\text{diag}}}$ is mainly determined by the survival probabilities of the two species, that is, Li–DMC complex and free DMC, as well as the conditional probability due to chemical exchange process from Li–DMC to free DMC. Thus, this intensity ratio is in fact quantitatively a good measure of the chemical exchange rate[33].

From the 2D Gaussian fit to the 2D infrared spectra (at 10 ps) for 1.0 M LiPF₆/DEC solution shown in Fig. 3c, we found that the intensity ratio, which is a dimensionless quantity, is 0.67. Now, for the 2D infrared spectra (10 ps) for 1.0 M LiPF₆/DMC:PC (1:1) solution in Fig. 5, our estimated intensity ratio is 0.76. In the cases of the 1.5 M LiPF₆/DMC:PC (1.5:1) and 2.0 M LiPF₆/DMC:PC (1.5:1) solutions, the intensity ratios are found to be 0.69 and 0.71, respectively (Supplementary Figs 10–12). Our observation that the ratio of cross peak intensity to the geometric mean value of the two diagonal peak intensities does not strongly depend on the solvent composition suggests that the chemical exchange timescale in LiPF₆/DEC solution is quantitatively similar to that in mixed solvents consisting of DMC and PC.

**Molecular dynamics simulations.** To examine the structures and dynamics of Li⁺ and PF₆⁻ solvation shells and ion aggregates, we carried out classical MD simulations for 1.0 M LiPF₆/DEC solution using the force field parameters for PF₆⁻ in refs 37,38 with the general Amber force field parameters for DEC. A representative snapshot structure of the solution is shown in Fig. 6a, where Li⁺ is represented by a yellow sphere and PF₆⁻ is by an octahedron. At this high lithium salt concentration, ions form not just CIPs but also large aggregates with a polydisperse distribution. Figure 6b depicts the configuration of ions only. Interestingly, certain large-size aggregates adopt ion network structures where lithium cations and hexafluorophosphate anions form spatially extended multi-branch chains and even three-dimensional network-like structures. We have recently found that cations and anions in high salt

aqueous solutions of KSCN, NaClO$_4$ and NaBF$_4$ show a strong propensity to form a large-scale spatially extended networks that are tightly intertwined with water H-bonding networks[39,40]. The fact that Li$^+$ and PF$_6^-$ ions form three-dimensional ion networks in LIB electrolyte solutions could be a key to understand detailed mechanism of Li$^+$ ion mobility, since Li$^+$ can jump from one to the other neighbouring anions in the same or different ion networks in the presence of non-zero gradient of electric potential[41]. Currently, we are carrying out a series of spectral graph analyses of the MD trajectories obtained for a variety of LIB electrolyte solutions and will present the results elsewhere since it is beyond the scope of this report.

## Discussion

Ultrafast fluxional changes of Li$^+$ solvation sheath structures in LIB electrolyte solutions are for the first time observed here with employing chemical exchange 2D infrared spectroscopic method. Combining FTIR spectroscopy, infrared pump-probe measurement and quantum chemistry calculation results, we completely characterized the vibrational properties of free DEC and Li–DEC complexes in both pure DEC solvent and mixed solvents consisting of DEC and PC. Subsequent 2D infrared data analyses revealed the ultrafast nature of chemical exchange dynamics in the immediate vicinity of Li$^+$ in DEC solution. Here one of the most difficult challenges was to prepare very thin solution samples because the infrared probes are the C$=$O stretch modes of DEC and PC molecules that are also solvents. Using RF magnetron sputtering technique, we could control the thickness of coated SiO$_2$ layer on CaF$_2$ window for linear and nonlinear infrared spectroscopic studies. The picosecond Li–DEC complex making and breaking dynamics in both pure DEC solution and DMC:PC mixed solution suggest that the macroscopic Li transport during both charging and discharging processes might be interrelated to microscopic solvation fluctuation and discrete transitions between different solvation states of Li$^+$ in LIB electrolyte solutions. We anticipate that the present ultrafast chemical exchange dynamics should also play an important role in solvation and de-solvation of Li$^+$ at the solid-electrolyte interface on electrodes because of the following reason. If the rigidity of each Li$^+$ solvation sheath is very high, any de-solvation of Li$^+$ on the anode surface may not be efficient, which potentially restricts the function of LIB due to lowering the rate of lithium deposition on the anode. On the other hand, if the Li$^+$ solvation sheath is not sufficiently stable due to a low dissolving power of solvent, the rate of Li$^+$ release from the cathode could be slow. Therefore, it is naturally expected that ideal LIB solvent molecules should interact with Li$^+$ with a properly balanced strength. Overall, the unexpected ultrafast fluxional solvation processes around Li$^+$ in the LIB electrolyte solution is turned out to be prerequisite for optimally achieving the functional role of carbonate solvent in LIB. We anticipate that a further 2D infrared investigation on the Li-solvent fluctuation dynamics during both charging and discharging processes would provide critical information on the fluctuation-dissipation relationship between equilibrium microscopic solvent fluctuation and non-equilibrium macroscopic charge transport phenomenon.

## Methods

**Sample preparation for infrared measurement.** Lithium hexafluorophosphate (LiPF$_6$, Battery grade, purity > 99.99% trace metal basis) and DEC (purity > 99%) was purchased from Sigma-Aldrich. DEC was dried using thermally activated 3 Å molecular sieves. The water content of the sample determined by Karl-Fisher method was less than 50 p.p.m. All solution electrolyte preparation and injection into the infrared cell holder was carried out in Ar-filled glove box (moisture < 1 p.p.m., oxygen < 20 p.p.m.). The low water content in sample for infrared absorption measurement was confirmed by examining the water O–H stretch band at around 3,500 cm$^{-1}$ in the infrared spectra.

In the present work, the carbonyl stretch mode of DEC was used as the infrared probe for monitoring any local change and dynamics of Li-solvation sheath. Since the infrared chromophore, DEC, is also solvent, the transmittance of infrared beam is very low so that an infrared sample holder with extremely narrow path length was needed. Previously, to overcome a similar difficulty, Cowan et al. used a ultra-thin sample cell consisting of 2 mm $\times$ 2 mm window of 800 nm-thick Si$_3$N$_4$ for 2D infrared study of pure water, where the effective thickness of water flow channel was controlled to be 500 nm (ref. 42). Here to make an infrared sample cell and holder for the commercially available FT-IR spectrometer and subsequent 2D infrared measurements, we used SiO$_2$ to coat on the 25 mm diameter CaF$_2$ window (Fig. 1a). Here SiO$_2$ was selected because it is chemically non-reactive and also it is strongly adhesive to CaF$_2$. RF (radio frequency) magnetron sputtering system (Supplementary Note 1) was used with three-inch diameter high-purity SiO$_2$ target. The thickness of sputtered SiO$_2$ layer on CaF$_2$ surface was verified by scanning electron microscopy and used the linear relationship between the sputtering time and the thickness of SiO$_2$ layer (Fig. 1b). In the infrared pump-probe and 2D infrared measurements, to minimize the re-absorption of the generated third-order infrared signal electric field, that is, self-attenuation problem, the absorbance of DEC carbonyl stretching mode was adjusted to be less than 0.4, which was achieved by using 800 nm path length. All the infrared spectra were measured on Jasco 6300 FV spectrometer with a circulating water bath for temperature control. This spectrometer is equipped with a DLATGS detector with Peltier element in the absorbance mode range from 450 to 7,800 cm$^{-1}$ with frequency resolution of 0.07 cm$^{-1}$. During the measurements, N$_2$ gas is purged to suppress any vapour noise and to minimize undesired water condensation on the CaF$_2$ window.

**Polarization selective infrared pump-probe and 2D infrared spectroscopy.** Experimental details of pump-probe and 2D infrared spectroscopy employed in the present study have been described elsewhere. Briefly, 800 nm pulses with duration of ~100 fs and 1.0 mJ were generated from Ti:Sapphire oscillator and regenerative amplifier operating at 1 KHz, which were then used to produce near- infrared pulses at ~1.4 and ~1.9 μm. With AgGaS nonlinear optical crystal, the signal and idler pulses are mixed to generate mid-infrared pulses with ~8 μJ per pulse and ~100 fs duration centred at 2,070 cm$^{-1}$. For polarization-selective infrared pump-probe measurements, wire grid polarizers were used to set the polarization directions of the pump and probe beams to be 45° and 0° before the sample. The probe beam was resolved after the sample to either +45° (parallel) or −45° (perpendicular) by a linear polarizer on a computer-controlled rotator, sent through a polarizer fixed to 0° into the spectrograph, and detected by a 32-element array detector.

**MD simulation method.** Molecular dynamic (MD) simulations were performed for LiPF$_6$-DEC solutions at 1.0 M LiPF$_6$ concentration, where 378 LiPF$_6$ molecules were dissolved in 1,000 DEC molecules. The force field parameters developed by Lopes and Pádua[40,41] and the general Amber force field parameters were used to describe PF$_6^-$ ion and DEC molecule. respectively, where the corresponding restrained electrostatic potential charges were obtained by carrying out B3LYP/6-311$++$G(3df,2pd) calculations. The particle mesh Ewald method was used for long-range electrostatic interaction and the cutoff distance for nonbonding interaction was set to be 10 Å. The solution system was first energy-minimized with the steepest descent method and the conjugate gradient method before running equilibrium MD simulations. Subsequently, a constant $N$, $p$ and $T$ ensemble simulation at 1 atm and 298 K was carried out for 2 ns to adjust the solution density. An additional 10 ns $N$, $V$ and $T$ ensemble simulation at 298 K was performed to make each LiPF$_6$-DEC solution system to reach its thermal equilibrium state. Finally, the production run was performed for 10 ns at constant $N$, $V$ and $T$ conditions, where the simulation time step was set to be 1 fs, and atomic coordinates were saved for every 1 ps.

**Data availability.** The data that support the findings of this study are available from the authors on reasonable request, see author contributions for specific data sets.

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

## Acknowledgements

This work was supported by IBS-R023-D1. K.-K.L. thanks Dr Ho-Jung Sun (Kunsan National University) for allowing us to use the sputtering system and for insightful discussion.

## Author contributions

M.C., K.-K.L. and K.K. designed the research. K.-K.L., K.P., Y.N. and D.K. performed the FTIR, infrared pump-probe and 2D infrared measurements. H.L. carried out density functional theory calculations. M.C. and K.K. wrote the manuscript.

## Additional information

**Competing financial interests:** The authors declare no competing financial interests.

