## [Peer Review File · Nature Communications]

Reviewers' comments:

Reviewer #1 (Remarks to the Author):

This is an interesting work on the dynamical properties of the first solvation shell of the Li⁺ ion in diethyl carbonate (DEC). The information obtained from ultrafast and two-dimensional vibrational spectroscopic methods is important for obtaining insight into what is going on in solutions.

There are a few comments on the content of this manuscript. The manuscript should be revised on these points before it is accepted for publication.

1.

On page 3, the authors state that the preferential solvation of Li⁺ in mixed solvent was confirmed by electron spray ionization mass spectroscopy (ESI-MS). This is not sufficiently correct. In the ESI-MS measurements, some molecules are possibly stripped off from the solvation shell in the detection process, so that the number of solvents derived from those measurements tends to be much smaller (~2) than expected (~4). In addition, ref 21 that the authors cite in this relation is on the 17O NMR study, not on the ESI-MS study.

2.

On page 5, the authors discuss the vibrational frequency shifts of the C=O stretching band of DEC induced by the interaction with the Li⁺ ion. The magnitudes of the frequency shifts derived from the calculations (~ 100 cm⁻¹), however, are much larger than that shown in Figure 1c.

One possible reason for this is that only one DEC molecule is included in the calculations. The magnitude of the vibrational frequency shift depends on the number of solvent molecules surrounding the ion. The authors should have noticed this, because they have done such calculations as shown in Figure S4.

In this relation, I also wonder why the conformation of DEC is different between Figure 2 on one hand (trans) and in Figures 4 and S4 on the other (one cis). I know that an isolated DEC molecule has the trans conformation, but as it interacts with Li⁺, it tends to have the cis conformation because of the larger dipole moment (related to the direction of the O-C bond). What is important is that the C=O stretching frequency also depends on the conformation, as shown in Figure S4. Therefore, it is preferable to include the conformation dependence in the discussion on page 5.

3.

In the present work, the authors study the dynamical properties of the first solvation shell of Li⁺ in DEC, not in a mixed solvent. As the authors state on page 4, DEC comparatively weakly interacts with Li⁺. Therefore, a highly mobile nature is expected. I wonder how it is related to the dynamics of mixed solvents, which are actually used in LIBs.

4.

In the Supplementary Information, on page 14, the authors state that [in JPCL 6, 3296 (2015)] no direct spectral feature indicating (such) direct interaction of DEC with Li⁺ in mixed solvent system was observed. This is incorrect. In that work, the Raman spectra in the region around 900 cm⁻¹ were also observed, and the observed spectral changes clearly indicate the direct C=O...Li⁺ interaction.

Reviewer #2 (Remarks to the Author):

The manuscript employs a vibrational spectroscopy 2D-IR to investigate the in-situ solvation sphere

dynamics of Li-ion in non-aqueous electrolytes. This topic itself is very important in understanding how Li-ion transport within the bulk electrolyte, how the solvated Li-ion becomes desolvated/solvated at the interfaces, and even how the so-called SEI is formed initially when pristine electrode comes in contact with electrolyte under electric field. However, the content of this article only establishes the spectroscopy as a valid technique in identifying the pico-seconds exchange rate of the solvation sphere members, which is not actually new knowledge. Therefore I think the current format does not support its publication at Nat Comm. A specialized journal seems to be more appropriate. On the other hand, if the manuscript contains extra results closer to real battery electrolytes, such as the relative exchange rates of different solvents in a multiple-solvent electrolyte, I am willing to reconsider my stance regarding this manuscript's qualifications.

REVIEWERS' COMMENTS:

Reviewer #1 (Remarks to the Author):

Reading the revised version of the manuscript, I think it is almost sufficiently revised.

There is only one question:

What is the relation between newly-added Figure 5 and Figure S10? These are the 2D-IR spectra of the same solution, in the same frequency region, but they look slightly different.

Reviewer #2 (Remarks to the Author):

The authors have added a lot new experiments and data. Most importantly, they investigated the mixed solvent system, and showed that while linear carbonate DMC adopts similar exchange behavior as DEC, the cyclic carbonate PC is relatively stable within the solvation shell. One would guess similar static nature from EC. This observation and the quantitative exchange rate data are every important for the developer of new electrolyte systems. As stated in my first round review comments, I am now ready to change my stance on the qualification of this paper. I recommend its publication.

Reviewer #3 (Remarks to the Author):

Review of MS #NCOMMS-16-17574A by Lee et al.

This manuscript reports on a 2DIR study of the exchange dynamics in the Li⁺ solvent sheath of Li-electrolyte solutions in carbonate solvents. The experiments and data analysis have been performed with care, and the results, which are explained in a clear and concise manner, are interesting and will have a broad scientific impact, which makes this paper very suitable for ncomms.

The authors did an excellent job in addressing the comments of reviewers #1 and #2. It may be noted that reviewer #2's comment that the manuscript "only establishes the spectroscopy as a valid technique in identifying the pico-seconds exchange rate of the solvation sphere members, which is not actually new knowledge" is incorrect: the main message of this paper is not about the technique, but about the results that were obtained with it.

Hence, this manuscript is definitely suitable for publication in ncomms. However, the authors should address the following minor issues:

1. The authors are very open about the contribution of local heating to their signals, in particular in their discussion of the vibrational and orientational lifetimes. However, it appears that in the analysis of the waiting-time dependent 2DIR spectra this local-heating contribution is not considered. Given that the local-heating contribution looks rather similar to a 'real' pump-probe signal (i.e. caused by vibrational-excited-state population), as can be seen in fig.3b, this potential local-heating contribution to the 2DIR signal should be discussed. For longer and longer waiting times, one would expect the heating contribution to eventually dominate the 2DIR spectrum. Can the authors exclude that the 2DIR spectra with the longest T_w are mainly temperature-difference-2DIR spectra? If the 2DIR signal at long T_w is still due to the vibrational-excited state population, then its overall amplitude should still decrease exponentially with T_w (with time constant = the vibrational lifetime). Unfortunately it cannot

be seen from the graphs in fig.3c or the SI whether this is the case.

2. Uncertainties in the reported time constants should be given. For instance, the reported value of 1.48ps for the rotational correlation time constant of free DEC suggests a precision that seems a bit unrealistic.

3. Finally, when introducing 2DIR exchange spectroscopy, a number of literature references are given (references 22-26). The authors may wish to also include here a reference to Chem. Phys. 266, 137 (2001), which was the first study to propose and demonstrate the idea of 2DIR-exchange spectroscopy.

Reviewer #1 (Remarks to the Author):

This is an interesting work on the dynamical properties of the first solvation shell of the Li⁺ ion in diethyl carbonate (DEC). The information obtained from ultrafast and two-dimensional vibrational spectroscopic methods is important for obtaining insight into what is going on in solutions.

There are a few comments on the content of this manuscript. The manuscript should be revised on these points before it is accepted for publication.

Comment 1. On page 3, the authors state that the preferential solvation of Li⁺ in mixed solvent was confirmed by electron spray ionization mass spectroscopy (ESI-MS). This is not sufficiently correct. In the ESI-MS measurements, some molecules are possibly stripped off from the solvation shell in the detection process, so that the number of solvents derived from those measurements tends to be much smaller (~2) than expected (~4). In addition, ref 21 that the authors cite in this relation is on the 17O NMR study, not on the ESI-MS study.

Reply 1. The reviewer is correct that ref 21 cited in the original manuscript is on NMR study not on the ESI-MS study. Now, it is properly corrected in the revised manuscript. We thank the reviewer.

Comment 2. On page 5, the authors discuss the vibrational frequency shifts of the C=O stretching band of DEC induced by the interaction with the Li⁺ ion. The magnitudes of the frequency shifts derived from the calculations (~ 100 cm⁻¹), however, are much larger than that shown in Figure 1c. One possible reason for this is that only one DEC molecule is included in the calculations. The magnitude of the vibrational frequency shift depends on the number of solvent molecules surrounding the ion. The authors should have noticed this, because they have done such calculations as shown in figure S4. In this relation, I also wonder why the conformation of DEC is different between figure 2 on one hand (*trans*) and in figures 4 and S4 on the other (*cis*). I know that an isolated DEC molecule has the *trans* conformation, but as it interacts with Li⁺, it tends to have the *cis* conformation because of the larger dipole moment (related to the direction of the O-C bond). What is important is that the C=O stretching frequency also depends on the conformation, as shown in figure S4. Therefore, it is preferable to include the conformation dependence in the discussion on page 5.

Reply 2. The reviewer is correct. Indeed, the C=O stretch frequency depends on the conformation of side chain that can adopt either *trans* or *cis* form. The structures shown in Figure 2 are just two representative ones considered only for the quantum chemistry calculations with varying distance between lithium ion and DEC molecule. As pointed out by the reviewer, the side chains of DEC in complexes with more than one DEC molecules or in those complexes in solution can adopt *cis* form too. To examine detailed structures of DEC molecules in solutions, we have additionally carried out MD simulations (new figure 6). We thank the reviewer for letting us know that the dipole moment of DEC in the *cis* conformation is larger than that in the *trans* conformation. As a consequence, the *cis* conformer can form a

stronger interaction with lithium ion and thus the C=O frequency depends on the side chain conformation. On page 5 of revised manuscript, such conformation dependence of the C=O frequency is discussed. Later in the revised manuscript, we present new discussions on our MD simulation results with a particular emphasis on the ion aggregation formation.

Comment 3. In the present work, the authors study the dynamical properties of the first solvation shell of Li⁺ in DEC, not in a mixed solvent. As the authors state on page 4, DEC comparatively weakly interacts with Li⁺. Therefore, a highly mobile nature is expected. I wonder how it is related to the dynamics of mixed solvents, which are actually used in LIBs.

Reply 3. This is an important question. Over the last three months, we put forth our efforts to carry out new 2D IR experiments on mixed solvents that are (i) 1.0 M LiPF₆ in DMC:PC (=1:1 in volume percent) solution, (ii) 1.5 M LiPF₆ in DMC:PC (= 1.5:1) solution, and (iii) 2.0 M LiPF₆ in DMC:PC (= 1.5:1) solution. Note here that we used DMC (dimethyl carbonate) instead of DEC. The reason why we chose DMC:PC mixed solvents instead of DEC:PC mixed solvents is two-fold. First, since the C=O stretching vibrational lifetime of DMC is comparatively longer than those of DEC and PC, we could observe measurable diagonal and cross peaks of free DMC and Li-DMC in the 2DIR spectra at long waiting times. Second, it is shown that the chemical exchanges observed in Li-DEC solution do occur in different mixed solvent systems too.

Despite that the 2D IR spectra appear to be highly spectrally congested, we could clearly observe chemical exchanges between Li-DMC complex and free DMC molecules even in DMC+PC (1.5:1 in volume percent) mixed solvents with 1.5 M LiPF₆ (see new figure 5 in the revised manuscript as well as figures S10-S12 in revised Supplementary Information and then compare the 2D IR spectra (at times 6 and 10 ps) in figure 3c with those (in red boxes) in figure 5). We found that the timescale of chemical exchanges in the lithium solvation shell remains approximately the same even in the case of the (DMC:PC) mixed solvents. Detailed discussions on quantitative analysis results for lithium electrolyte solutions with mixed solvents is newly added to the revised manuscript (on pages 11 and 12) as well as revised Supplementary Information (figures S10-S12 and sections VI and VII newly added to the revised Supp. Info.).

Comment 4. In the Supplementary Information, on page 14, the authors state that [in JPCL 6, 3296 (2015)] no direct spectral feature indicating (such) direct interaction of DEC with Li⁺ in mixed solvent system was observed. This is incorrect. In that work, the Raman spectra in the region around 900 cm⁻¹ were also observed, and the observed spectral changes clearly indicate the direct C=O...Li⁺ interaction.

Reply 4. We thank the reviewer very much for pointing out our incorrect referencing. This is corrected now in the revised manuscript.

Reviewer #2 (Remarks to the Author):

The manuscript employs a vibrational spectroscopy 2D-IR to investigate the in-situ solvation sphere dynamics of Li-ion in non-aqueous electrolytes. This topic itself is very important in understanding how Li-ion transport within the bulk electrolyte, how the solvated Li-ion becomes desolvated/solvated at the interfaces, and even how the so-called SEI is formed initially when pristine electrode comes in contact with electrolyte under electric field. However, the content of this article only establishes the spectroscopy as a valid technique in identifying the pico-seconds exchange rate of the solvation sphere members, which is not actually new knowledge. Therefore I think the current format does not support its publication at Nat Comm. A specialized journal seems to be more appropriate.

On the other hand, if the manuscript contains extra results closer to real battery electrolytes, such as the relative exchange rates of different solvents in a multiple-solvent electrolyte, I am willing to reconsider my stance regarding this manuscript's qualifications.

Authors' response. In fact, the reviewer 1 also made the same comment “*In the present work, the authors study the dynamical properties of the first solvation shell of Li+ in DEC, not in a mixed solvent. As the authors state on page 4, DEC comparatively weakly interacts with Li+. Therefore, a highly mobile nature is expected. I wonder how it is related to the dynamics of mixed solvents, which are actually used in LIBs.*” We thank both reviewers for pointing out the limitation of our previous experimental results on LiPF₆ in DEC solvent, which is different from the more realistic mixed solvent systems.

Over the last three months, we put forth our efforts to carry out new 2D IR experiments on mixed solvents that are (i) 1.0 M LiPF₆ in DMC:PC (=1:1 in volume percent) solution, (ii) 1.5 M LiPF₆ in DMC:PC (= 1.5:1) solution, and (iii) 2.0 M LiPF₆ in DMC:PC (= 1.5:1) solution. Here, it should be noted that we used DMC (dimethyl carbonate) instead of DEC. The reason why we chose DMC:PC mixed solvents instead of DEC:PC mixed solvents is two-fold. First, since the C=O stretching vibrational lifetime of DMC is comparatively longer than those of DEC and PC, we could observe measurable diagonal and cross peaks of free DMC and Li-DMC in the 2DIR spectra at long waiting times, otherwise the 2DIR spectra would be completely dominated by the intense peaks from PC molecules and Li-PC complexes with longer lifetimes. Second, we want to show that the chemical exchanges observed in Li-DEC (single carbonate solvent) solution do also occur in different alkyl carbonate mixed solvent systems.

Despite that the 2D IR spectra appear to be highly (spectrally) congested (see newly added figure 5 in the revised manuscript), we could clearly identify a cross peak at $\omega_\tau = 1725 \text{ cm}^{-1}$ and $\omega_i = 1760 \text{ cm}^{-1}$, which evidences chemical exchanges (transitions) between Li-DMC complex and free DMC molecules even in DMC:PC (1.5:1 in volume percent) mixed solvents with 1.5 M LiPF₆. Such chemical exchanges are also observed in other mixed solvents with different components (see figures S10-S12 in the revised Supplementary Information). We found that the timescale of the chemical exchanges in the lithium solvation shell remains approximately the same even in the case of the (DMC:PC) mixed solvents. Detailed discussions on quantitative analysis results for lithium electrolyte solutions with mixed solvents have been newly added to the revised manuscript (on pages 11 and 12) as well as the revised Supplementary Information (figures S10-S12 and sections VI and VII

newly added to the revised Supp. Info.).

To understand solvation structures, we further carried out molecular dynamics simulations (see pages 12 and 13 in revised manuscript and new figure 6 depicting a representative snapshot structure of 1.0 M LiPF_6 in DEC solvent) and provide discussions that the formation of polydisperse ion aggregates could be a key to understand solvation dynamics in lithium electrolyte solutions.

We now hope that the reviewer 2 finds the revised manuscript interesting and publishable in Nature Comm.

AUTHORS RESPONSES TO REVIEWERS' COMMENTS

Reviewer #1 (Remarks to the Author):

Comment. Reading the revised version of the manuscript, I think it is almost sufficiently revised. There is only one question: What is the relation between newly-added Figure 5 and Figure S10? These are the 2D-IR spectra of the same solution, in the same frequency region, but they look slightly different.

Reply. We thank the reviewer 1 for pointing out this. We made a correction to the figure 5 by replacing it with that in supplementary figure 10-a.

Reviewer #2 (Remarks to the Author):

Comment. The authors have added a lot new experiments and data. Most importantly, they investigated the mixed solvent system, and showed that while linear carbonate DMC adopts similar exchange behavior as DEC, the cyclic carbonate PC is relatively stable within the solvation shell. One would guess similar static nature from EC. This observation and the quantitative exchange rate data are every important for the developer of new electrolyte systems. As stated in my first round review comments, I am now ready to change my stance on the qualification of this paper. I recommend its publication.

Reply. We are very glad to hear that the reviewer 2 finds our work interesting and our manuscript publishable.

Reviewer #3 (Remarks to the Author):

Comments. This manuscript reports on a 2DIR study of the exchange dynamics in the Li⁺ solvent sheath of Li-electrolyte solutions in carbonate solvents. The experiments and data analysis have been performed with care, and the results, which are explained in a clear and concise manner, are interesting and will have a broad scientific impact, which makes this paper very suitable for ncomms.

The authors did an excellent job in addressing the comments of reviewers #1 and #2. It may be noted that reviewer #2's comment that the manuscript "only establishes the spectroscopy as a valid technique in identifying the pico-seconds exchange rate of the solvation sphere members, which is not actually new knowledge" is incorrect: the main message of this paper is not about the technique, but about the results that were obtained with it.

Hence, this manuscript is definitely suitable for publication in ncomms. However, the authors should address the following minor issues:

Comment 1. The authors are very open about the contribution of local heating to their signals, in particular in their discussion of the vibrational and orientational lifetimes. However, it appears that in the analysis of the waiting-time dependent 2DIR spectra this local-heating contribution is not considered. Given that the local-heating contribution looks rather similar to a 'real' pump-probe signal (i.e. caused by vibrational-excited-state population), as can be seen in fig.3b, this potential local-heating contribution to the 2DIR signal should be discussed. For longer and longer waiting times, one would expect the heating contribution to eventually dominate the 2DIR spectrum. Can the authors exclude that the 2DIR spectra with the longest T_w are mainly temperature-difference-2DIR spectra? If the 2DIR signal at long T_w is still due to the vibrational-excited state population, then its overall amplitude should still decrease exponentially with T_w (with time constant = the vibrational lifetime). Unfortunately it cannot be seen from the graphs in fig.3c or the SI whether this is the case.

Reply 1. We appreciate the reviewer's valuable comment on a possible contribution from the local heating to the 2D-IR spectrum. The reviewer is correct that the local heating contribution needs to be considered in quantitatively analyzing 2D-IR spectra at longer waiting times. In fact, we also studied how such local heating affects on cross and diagonal peaks in 2D-IR spectra (*Ultrafast intermolecular vibrational excitation transfer from solute to solvent: Observation of intermediate states*, H. Son, K.-H. Park, K.-W. Kwak, S. Park, and M. Cho, *Chem. Phys.* **422**, 37-46, 2013). However, in the present, it is quite fortunate that such long-time local heating effect does not make much difference in estimating the exchange rate constants from our 2D-IR spectra because the timescales of chemical exchange processes in the present case of Li-ion electrolyte solutions are much faster than that of local heating process. That is to say, the 2D-IR spectra taken into consideration for estimating chemical exchange time constants are not dominated by the local heating effect. In our pump-probe measurements and analyses, we took data up to 30 ps (supplementary figure 6 shows the full trajectory of pump-probe experiment). However, to avoid any complexity resulting from such local heating effect, we considered the 2D-IR spectroscopic data only up to 10 ps for detailed kinetic analyses as shown in figure 4, even though we collected 2D-IR data until 14 ps and longer (see supplementary figure S).

As shown in figure 3a and 3b (pump-probe data), the 0-1 transition (positive) signals extend to a significantly longer waiting time and keep on decaying until 30 ps (see supplementary figure 6) instead of rising (in-growing) or reaching an asymptotic constant value. Furthermore, the negative 1-2 (excited state absorption) peaks almost decay to zero in ~ 10 ps as shown in figure 3a. Thus, we strongly believe that the experimental data up to waiting times of ~ 10 ps are mainly determined by chemical exchange and vibrational relaxation dynamics instead of local heating. Nonetheless, we shall further investigate this and many other related Li electrolyte solutions to develop kinetic models that are applicable to these systems.

Comment 2. Uncertainties in the reported time constants should be given. For instance, the reported value of 1.48ps for the rotational correlation time constant of free DEC suggests a precision that seems a bit unrealistic.

Reply 2. We have added proper error ranges to all the measured vibrational and orientational relaxation time constants with fully taking consideration of significant figures.

Comment 3. Finally, when introducing 2DIR exchange spectroscopy, a number of literature references are given (references 22-26). The authors may wish to also include here a reference to Chem. Phys. 266, 137 (2001), which was the first study to propose and demonstrate the idea of 2DIR-exchange spectroscopy.

Reply 3. The reviewer is correct so that the reference mentioned above has been added to the revised manuscript as ref. 23.